# The Role of the Extracellular Matrix in Inducing Cardiac Cell Regeneration and Differentiation

**DOI:** 10.3390/cells14120875

**Published:** 2025-06-10

**Authors:** Nicla Romano

**Affiliations:** Department of Life Sciences, Health and Health Professions, Link Campus University, 00165 Rome, Italy; n.romano@unilink.it

**Keywords:** zebrafish, heart, extracellular matrix

## Abstract

The adult human heart has a limited ability to regenerate after injury, leading to the formation of fibrotic scars and a subsequent loss of function. In fish, mice, and humans, cardiac remodeling after myocardial injury involves the activation of epicardial and endocardial cells, pericytes, stem cells, and fibroblasts. The heart’s extracellular matrix (ECM) plays a significant role in the regeneration and recovery process. The epicardium, endocardium, and pericytes reactivate the embryonic program in response to ECM stimulation, which leads to epithelial–mesenchymal transition, cell migration, and differentiation. This review analyzes the role of ECM in guiding the differentiation or dedifferentiation and proliferation of heart components by comparing significant findings in a zebrafish model with those of mammals.

## 1. Introduction

Heart failure contributes to a large number of deaths. Heart failure includes cardiac structural or functional abnormalities that impair cardiac filling and output. In turn, this causes the development of cardiovascular pathologies, such as cardiac hypertrophy, aortic aneurysm, and lower extremity peripheral arterial disease. Thus, cardiovascular disease represents the main cause of death in humans worldwide. Approximately 20 million people die every year because of 18 risk factors, accounting for more than 32% of global deaths [1]. Myocardial infarction is caused by a decrease in or suspension of blood flow to a part of the heart, which leads to necrosis of the heart muscle and, therefore, the death of many cardiomyocytes, the muscular component of the heart [2]. After such an adverse episode, cardiac adverse remodelling includes biological changes that affect the composition and architecture of the extracellular matrix (ECM). The consequently disrupted signaling can interfere with the balance between cardiogenic and pro-fibrotic phenotypes of resident cardiac stromal cells [3]. Following the loss of functional cells after an infarction, the heart activates an inflammatory response that determines the recruitment of fibroblasts, which secrete numerous growth factors and matrix components. The activation of the inflammatory process provokes hypertrophy around the site of infarction and in some surrounding areas. In hypertrophic disease, the increased diameter of cardiomyocytes resulting from fibrillogenesis is accompanied by extracellular fibrosis due to matrix deposition, causing an increase in ventricular wall pressure. Thus, the two pathologies of infarction and hypertrophy are often linked to each other.

Growth factors secreted by fibroblasts can stimulate cardiac cells (endothelial and epicardial cells and cardiomyocytes) to secrete, in turn, factors that activate and balance the regeneration process [4,5,6]. This process is directed by the expression of specific microRNAs that control the expression of embryonic and differentiation genes [7,8]. A similar process seems to occur in the hypertrophic pathology of the heart, where the activation of fibroblasts can cause the deposition of copious quantities of matrix, impeding the elastic capacity of the organ itself [9,10,11,12]. In addition to clinical studies, researchers can use alternative laboratory models to study pathologies or enhance the intrinsic regenerative capacity of the heart. The human cardiosphere, organoids, and zebrafish represent the best models for studying the role of ECM in inducing cardiac cell fate. This review examines key factors in the extracellular matrix (ECM) that play a role in the regeneration and differentiation of heart tissue, as described in both zebrafish and mammalian models. This panel of information may be useful as a basis of knowledge to actuate new research strategies for heart repair.

## 2. Inducing Cardiac Cells via Key Morphogenic Factors

In the past, research into cell therapies was the only way to make possible progress in heart repair and regeneration in mammals. Several investigations have focused on bone marrow-derived stem cells or resident cardiac stem cells cultured in 3D as cardiospheres [13,14]. These studies assumed that exogenous cells delivered into the heart would transdifferentiate into functional cardiomyocytes and that endogenous cardiac stem cells would functionally couple with existing cardiomyocytes [15,16,17,18]. The success of these approaches has been limited given the poor survival of exogenous stem cells; therefore, the benefits are largely limited to short-term paracrine effects or the immune response triggered by the injected cells [19,20]. Recently, heart muscle cells differentiated in vitro from induced pluripotent stem cells (iPSCs) were successfully injected into a patient; however, only partial repair occurred due to the limited proliferation of these cells in situ [21]. In contrast, the zebrafish model has demonstrated a high capacity for proliferation and regeneration of resident cardiomyocytes, both in vivo and in ex vivo culture, without the need for stem cell injection [5,22]. Moreover, it has been demonstrated that it is possible to regenerate a heart ex vivo without the fish body by using the right cocktail of growth factors [5]. Therefore, insights from zebrafish regeneration mechanisms may be essential for applying similar approaches to mammalian model systems, helping to identify effective stimulation and repair strategies.

Adult mammalian hearts have a very low baseline cardiomyocyte renewal rate (~0.76% per year), which increases to about 15% following cardiac infarction. Despite this increase, the repair process typically results in scar formation rather than functional tissue regeneration, because the renewal rate remains insufficient for complete recovery [22,23,24,25]. Neonatal mammalian hearts exhibit a much higher regenerative potential, comparable to that of zebrafish; during this early period, mammals can regenerate heart tissue effectively, avoiding fibrosis and scar formation. Zebrafish are remarkable for their ability to regenerate the entire heart, including functional myocardium, within 30-50 days without forming scars or fibrosis, making them a valuable model for understanding heart regeneration mechanisms [26,27]. This regenerative capacity is driven by gene expression shared from fish to mammals and is induced by extracellular stimulation [28,29]. Therefore, a novel approach to supporting cardiac regeneration without stem cells could involve stimulating native cells, with the extracellular matrix (ECM) potentially playing a crucial role.

The morphogenesis of the heart and the maintenance of its physiology require the activation and coordination of different transcriptional programs, which are activated starting from the expression of homeobox genes and exocytosis morphogenetic factors such as Nodal, BMP, FGF, Wnt, and Notch. Cellular signalling pathways triggered by these factors play a significant role in cardiac specification and differentiation [30,31,32,33,34,35,36,37,38,39,40] (Table 1). A panel of detailed information on the morphogenetic and differentiation processes in early cardiac development divergences overlapping between zebrafish and human hearts is included in the review by Angom et al. [38]. In brief, studies on cardiac regeneration in zebrafish have highlighted the fundamental roles played by non-muscle cardiac tissues and cells such as the epicardium, endocardium, and fibroblasts [5,41,42,43,44,45]. The epicardium is a mesothelial layer that surrounds the heart in vertebrates [46]. Epicardial cells are derived from a cluster of transient cells of pericardial mesodermal origin in the embryo, induced by the descent of neural crest cells [46]. Following formation, the transient cells of the epicardium flatten and cover the developing heart to create two to three contiguous cell layers in zebrafish [47]. In cases of induction of cardiac hypertrophy generated by phenylephrine, there may be an increase to five or six layers [48]. Embryonic epicardial cells, which therefore maintain a high degree of potential, express certain gene markers, among which those of particular importance are GATA4, Wt1, Tbx18/20, Raldh2, and the hyaluronic acid receptor [5,48,49,50,51]. This gene activation instructs the epicardium to undergo a process of epithelial-mesenchymal transition (EMT), during which epithelial cells detach from the primary tissue and migrate into the subepicardial space until they invade the myocardial wall, where they probably give rise to myocyte epicardial derivation [32,45,52]. In fact, these transdifferentiated cells contribute to the formation of several cell lineages, including endothelial/smooth muscle cells of the coronary vascular network, endocardial mesenchymal cells of the atrioventricular valves, and cardiac fibroblasts [41,52,53,54]. During activation by the ECM, the activated/transdifferentiated epicardium re-expresses embryonal genes (i.e., GATA4, Wt1, Raldh2 and Tbx18) and starts the proliferation and migration processes of the cells themselves in the injured myocardium [8,55]. This activation has been observed in development as well as after infarction and in phenylephrine-induced hypertrophy [44,45,56,57]. The activation of embryonic pathways may also depend on high-mobility group box1 (HMG1A and HMG1B) proteins that are involved in DNA repair, transcription regulation in the nucleus, and also in the regulation of autophagic function in the cytoplasm after heart damage [57]. In the ECM, these molecules are included in the “alarmins” category and, in particular, in the DAMP typology (damage-associated molecular pattern). DAMPs are endogenous proteins released from damaged or dying cells that can activate the inflammatory system and the machinery of heart regeneration in zebrafish as well as in mammals [58,59]. The subsequent establishment of crosstalk among the epicardium, myocardium, fibroblasts, and immune cells is crucial in order to allow the correct development/regeneration of cardiac tissues [42,52]. The expression of epicardial markers in regenerating tissue coincides temporally with the vascularization of newly formed myocardial tissue and with the activation of other endothelial and endocardial tissue markers, such as NFAT2 [10,11,60,61]. The endocardium can be considered a specialized endothelium that forms the innermost layer of the heart wall [60]. After a ventricular lesion or in a hypertrophic model in zebrafish, the entire endocardial layer undergoes morphological changes and begins to express the enzyme Raldh2 (retinaldehyde dehydrogenase 2), which is important for triggering the process of cardiomyocyte proliferation and epithelial-mesenchymal transition [50,62]. Endocardial Notch signalling is also essential in regeneration for the proliferation of cardiomyocytes; in fact, its inhibition decreases the proliferation of myocardial cells [63]. Moreover, endocardial cells together with fibroblasts contribute to the production of collagen [42]. The role of endocardial cells has undoubtedly been underestimated; these cells originate from the same Flk1-positive progenitors as smooth muscle cells during heart development. During development, the smooth myoblasts derived from the same mesenchymal bud of the vessels start to differentiate into cardiomyocytes, displaying striate myofibrils [64]. Previous research has suggested that because they share the same progenitor, endocardial cells have the potential to generate cardiomyocytes [65]. In Scl−/− developing mice, endocardial cells activated the expression of transcription factors (Isl1, Tbx5, Nkx2.5, GATA6, troponin T) that led the cells to become muscular instead of having an endocardial lineage [64]. Interestingly, the Wt1, NFAT2, and GATA4 transcription factors seemed to be translated in an alternative translation process involving the RACK1 protein [66].

Mural cells or pericytes are present in mammals and have been localized in the peripheral portion of zebrafish hearts, in the epicardial layer [67,68]. These pericytes stabilize the circulation through heart vessels via physical and molecular interactions with adjacent endothelial cells. They also have contractile functions that can regulate blood flow. Mural cells expressing PDGFR-β (platelet-derived growth factor receptor beta), CD146 (melanoma cell adhesion molecule), and NG2 (chondroitin sulphate proteoglycan 4) can transdifferentiate in vivo in myofibroblasts during injury-induced fibrosis [61].

**Table 1 cells-14-00875-t001:** Common transcription (A) and morphogenetic (B) factors in zebrafish and mammalian cardiac regeneration.

	*Zebrafish vs. Mammals*	*Expression Place*
** *(A) Transcription Factors* **		
GATA4	Mesoderm-to-cardiac fate transition [8,38,41,48,49,55]	Epicardium, cardiomyocytes, endocardium
Wt1 a/b	Activate epicardial/pericardial cells and transdifferentiation [5,8,55]	Pericardium and Epicardium
NFAT2	activate endothelial and endocardial cells in prolipheration/transdifferentiation [5,10,11,60,61]	Endocardium and heart Endothelium
TBX 18/TBX 20	Differentiation of transdifferentiate cells [22,32,42]	Cardiomyocyte and epicardium
HAND2	Promotes cardiomyocyte differentiation [8,55]	Cardiac ventricle (right ventricle in mammals)
MEF2	Drives cardiomyocyte differentiation [38]	Endotelium, endocardium
TCF21	Epicardium prolipheration and differentiation [42,69]	Transdifferentiate mesenchimal cells/fibroblast
HMG1A/B	Activating the expression of genes involved in the regeneration, including Raldh2,Isl1 [58]	Epicardial and endocardial transdifferentiate cells
** *(B) Growth Factors* **		
FGFs	Stimulation of mesodermal/mesenchymal cardiocytes [70,71,72,73,74,75,76]	Resident fibroblasts, ECM, epicardial/pericardial cells
PDGFR-β	Stimulation of transdifferentiation of cardiocytes [8,55,77,78]	Resident fibroblasts, ECM
Wnt/antagonist of Wnt	Control of differentiation/proliferation of cardiomyocytes [37,38,39,40,45,79]	Pericardium, epicardium, endocardium/ECM
Shh	Proliferation of mesenchymal cell, cardiomyocyte development [36,46,79]	Pericardium, epicardium
TGF-β	Fibroblasts stimulation to produce GFs and ECM elements after injury [79,80]	Epicardium, endocardium, endothelium
IGFs	Proliferation of cardiomyocytes after a damage [81,82]	Epicardium
BMP	Ventral mesodermal gradient manteinance [30,38]	Pericardium
NRG-1	Proliferation of cardiomyocytes after a damage; Macrophage stimulation [83,84,85]	Endothelial cells and endocardium in regeneration
FlK1	Differentiation in cardiomyocyte [43,64]	Endothelial cells and endocardium in regeneration

## 3. Inducing Cardiac Cells with Key Extracellular Components

Questions that have arisen in this field include “Which cells are induced in the cardiac parenchyma?” and “What are the inducing factors in the extracellular environment that lead to the expression of embryonal genes?”.

### 3.1. Cardiac Cell and Their Basal Lamia/ECM Adhesions

Cardiac epithelial cells are components of the epicardium, endocardium (also “pericytes” in mammals), and endothelium. These cells can transdifferentiate and proliferate, facilitating heart recovery from fish to mammals [86,87]. Cardiomyocyte autograph or xenograph implantations have demonstrated poor success using stem cells derived from cardiac niches or iPSC [86]. Resident stem cells have been identified in specific niches within the heart, playing a fundamental role in maintaining cardiac homeostasis and facilitating myocardial repair after injury [87]. These niches are located in specific areas (close to supporting cells/pericytes and endocardium), representing specialized microdomains where the quiescent and activated state of the resident stem cells is regulated [87]. Both resident and migrating cellular components, such as fibroblasts, resident macrophages, and cardiomyocytes, are involved in this activation (Figure 1).

The principal activation mechanism of these cells involves integrins, cadherins, cellular receptors, and chemical factors that are specific to each cell type [16,43]. Together, these mechanisms orchestrate the activation, communication, and functional responses of different cardiac cell types, such as cardiomyocytes, fibroblasts, endothelial cells, and immune cells, during development, homeostasis, and repair processes. The following discussion describes some recent and also some older insights into the role of ECM–cellular integrin/cadherin interactions in inducing cardiac tissue.

Integrins are the major adhesion receptors, signaling across the plasma membrane in both directions. The consequence of this adhesion is the establishment of a large molecular network among the components of the basal lamina (fibronectin, collagens of I and IV types, entactin, etc.) capable of inducing signal transduction in cells. For example, under adhesion stimulus, the transduction signaling can involve the RAS_ERK or RHO systems, which can lead the cell to expose new membrane molecules such as cadherins or other migrating integrins [88]. Integrins consist of two chains of α and β types. However, each chain can be variable in composition, depending on the domain’s constitution. The α chain contains multiple domains, with the αA domain featuring four subunits (α1, α2, α10, and α11) that specifically link to the β chain. The β1 subunit forms a distinct laminin/collagen-binding subfamily [89,90]. Integrins are linked on the cytoplasmic side to actin fibers by α-actin, vinculin, paxillin, and talin, and also by adaptor proteins Cas, Crk, Grg2, Src, and CSK, along with signaling molecule FAK (focal adhesion kinase) [91,92]. Integrins are responsible for the formation of focal complexes, which represent temporary adhesions to the ECM [93]. The focal adhesions represent activation of a signaling cascade that could further generate contractile force and cell migration [94]. Cell surface receptors can synergize with other cell surface receptors, such as growth factor receptors, to activate signaling pathways that affect the cell cycle, proliferation, and migration. This finding highlights the essential role of cell adhesion via integrins in the G1/S checkpoint transition and the successful completion of cytokinesis [89]. Integrins are also involved in differentiation, apoptosis, cell pathology, cell shape, trans-differentiation, and migration [95]. The activation of integrins can occur either through ligand binding or due to effects on the cytoplasmic domain, leading to the straightening and separation of the intramembrane/cytoplasmic tails [96]. The signal transduction pathways and many of the key players involved are associated with the activation of the Ras-ERK-Cyclin D pathway, the ROCK/MLCK-cytoskeleton pathway, and the PI3K-Akt pathway, leading to primary effects on cell behavior mediated by integrins, often acting in concert with G protein-coupled receptors or kinase receptors for soluble factors [89].

Cadherins are integral membrane proteins that become activated in cell-to-cell or cell-to-ECM adhesion in the presence of Ca2+-dependent homophilic interactions. Their role is to maintain the structural integrity of cardiac cells in the context of cell-cell adhesion and the migration of transdifferentiated cells during cardiac regeneration [5,87]. The role of cadherins in cardiac pathology has been studied in rats, revealing dysregulation of intercalated discs due to incorrect adherent junctions between myocardial cells [97], leading to myofibrillar aberrance [98]. Intercalated discs are essential structures unique to cardiac muscle; they facilitate mechanical coupling and chemical communication among adjacent cardiomyocytes, permitting the regulated contraction necessary for cardiac function [98]. To confirm this, experiments involving the knockout of cadherins in mice have revealed abnormal heart development and compromised fetal viability [99,100]. Furthermore, there is evidence of reduction in the expression of N-cadherin in the context of dilated cardiomyopathy [101]. In zebrafish with embryonic lethal heart failure, mutant sarcomere-contractile stress-responsive genes are severely downregulated due to mutations in integrin/cadherin receptors [101].

### 3.2. The Basal Lamina and ECM Components

The ECM components that interact with and activate integrins and cadherins include the arginyl glycyl aspartic acid (RGD) peptide motif, which is common in various ECM components (e.g., fibronectin, osteopontin, vitronectin, and fibrinogen) [90,102,103].

Fibronectin (FBN) is an ECM component that consists of two nearly identical protein monomers linked by disulfide bonds. FBN is assembled by cells into viscoelastic fibrils that can bind 40 distinct growth factors and cytokines. FBN binds to cellular integrins and ECM proteins such as collagen and fibrin, as well as heparan sulphate proteoglycans [103]. The LDV motif, which is contained in fibronectin, is recognized by α4β1, α4β7, and α9β1 integrin-ligand combinations [104,105]. Following integrin–LDV interaction, the fibronectin dimer undergoes structural changes, forming the fibrillar type of cell/ECM adhesion—one form of focal adhesion. This transition leads to a crucial state in which a provisional ECM is assembled during embryonic development and wound healing or, alternatively, contributes to hypertrophic pathological disease [93,94,106]. α5β1 integrin appears to be particularly involved in fibronectin–fibrillogenic formation [93], whereas FAK appears to be involved in the control of cell growth and migration in cadherin-mediated adhesions [107].

Laminins (LNs) >100 kDa are found primarily in the basal laminae [108]. They consist of three primary protein subunits (α, β, and γ) coiled together to form a supercoiled triple helix in 16 different combinations [109]. LNs do not form fibrils but organize themselves into reticular structures capable of resisting traction forces simultaneously in many directions [110]; their integrin binding sites (GOFGER) are present in all three chains. However, next to the integrin link site, numerous sites link collagen, entactin, perlecan, vitronectin, etc. Thus, the LN functions as “glue” between the basal lamina and the water-enriched matrix in the ECM. Entactin is one of the components of the basal lamina; it is a stabilizer of the link between laminin, type IV collagen, and cellular integrins [111]. When the molecule binds to calcium ions, it becomes active to perform a linking or bridging function [112].

Collagen is composed of repeating units of tropocollagen, a protein complex with a molecular mass of approximately 285 kDa, consisting of three α polypeptide chains. These α chains are encoded by 30 genes located across at least 12 different chromosomes, exhibiting slight variations in their amino acid sequences. This results in five distinct chains (labelled α1, α2, α3, α4, α5) that can variably associate to form a triple helix through hydrogen bonds. There are over 20 types of collagen, with variations arising from the assembly of tropocollagen monomers and the composition of the three α chains. Type I collagen consists of two α1 chains and one α2 chain with slightly different compositions, while type III collagen comprises one α1 chain and two α2 chains [113]. Additionally, type IV collagen is formed by the assembly of five different α chains and is anchored to the basement membranes of myocardial cells, making it essential for the elasticity of the contractile system [114]. Following an ischemic event and consequent myocardial necrosis, the heart undergoes remodeling and accumulation of collagen associated with the intense inflammatory process, called reparative fibrosis [115]. Fibroblasts produce procollagen, and the deposition of large amounts of new collagen has consequently been observed, leading the total cardiac mass to increase. In zebrafish, this process causes temporary hypertrophy, and the remodelling of tissues allows the heart to undergo reparative regeneration [22] even under ex vivo conditions [5]. In mammals, including humans, ischemia in the heart also brings about the activation of immunocytes and fibroblasts, but the permanent scarring of the affected area is accompanied by a consequent loss of contractility and pathologic hypertrophy. Reactive fibrosis is different from reparative fibrosis due to ischemic events; it demonstrates a diffuse deposition of collagen throughout the myocardium and has been studied mainly in mammals [116]. Collagen deposition is a critical process to study and regulate in order to prevent pathology, making it essential to monitor collagen levels in the extracellular matrix (ECM) [117]. Cardiac fibroblasts produce collagen I (C-I) and collagen III (C-III). Collectively, these proteins contribute to the fibrous meshwork, tissue alterations in ECM quality (e.g., changes in crosslinking), and changes in the proportion of components (including C-I to C-III) in the heart’s pathology [118]. Regulation of the amount and composition of the ECM occur in a dynamic process involving both fibroblast-C-I/C-III production and metalloprotease degradation [118]. Fibroblasts can be stimulated indirectly by the nervous system by stimulating the production of vascular and adrenal organ hormonal factors (i.e., angiotensin II, aldosterone, catecholamines). This induction pattern can be observed in patients with chronic stress, genetically predisposed obesity, diabetes, and metabolic syndrome [119,120,121]. Certain pathological inducing factors can stimulate fibroblasts, leading to excessive production of collagen types I and III or hypersecretion of growth factors.

### 3.3. Roles of Hyaluronan Acid and GAG

Special mention of hyaluronan acid (HA) should be made in relation to cardiac homeostasis and repair. HA has a polymeric structure of aggrecans and glycosaminoglycan composed of repeating polymeric glucuronic acid and up to 25,000 N-acetyl-glucosamine disaccharide repeats conjugated by glucuronic β fold bond linkages (GAG), with a total molecular weight of ~4000 kDa [122]. HA is abundantly expressed during tissue repair, supporting cells in relevant processes including survival, proliferation, and differentiation. However, it can also be overexpressed in pathological heart hypertrophy due to links with collagen hyper-production among cardiomyocytes [123]. Several studies have indicated that hyaluronan has a central role in regeneration in zebrafish, as well as in mammals [51,124]. In zebrafish, a proteomic study from a regenerating heart in vivo indicated that the HA-mediated motility receptor (which is fundamental in transdifferentiated cell migration) and HA synthases (to form hyaluronan acid) were highly expressed [51]. In mammals (rats), mesenchymal stem cells extracted from the placenta and patched onto the scar of the infarcted heart inhibited the hyperproduction of small-leucine-rich proteoglycan lumican, contributing to fibrosis [125]. Moreover, a heart grafted with HA-mixed ester-treated mesenchymal cells was able to normally produce mitochondrial respiratory enzymes and carbonic anhydrase-I instead of expressing the downregulation of these that typically occurs in a heart stroke. HA can modulate the crosstalk between growth factors and their receptors on cardiac cells. For example, it is well documented that PDGF-BB, TGF-β1, and CD44 signaling depend on the presence of hyaluronic acid [126]. In zebrafish, the PDGF signalling pathway is necessary to regenerate the heart in vivo [77,127], as well as ex vivo [5]. Moreover, the extracellular matrix (ECM) from a decellularized zebrafish heart—containing glycosaminoglycans (GAGs), hyaluronic acid, collagens, and growth factors—successfully regenerated an infarcted mouse heart [128].

### 3.4. Growth Factors in the Cardiac Stroma

Disease-specific pathophysiological perturbations may trigger distinct molecular patterns of fibroblast activation that modulate the composition of the interstitial ECM and the administration/diffusion of growth factors [129,130,131]. In turn, growth factors and ECM proteins can activate myofibroblasts and the epithelial cardiac components (Figure 1; Table 1). The environment and the cocktail of growth factors/proteins seem to be the key factors in inducing proliferation and regeneration or, conversely, cardiomyocytes’ dysfunctional properties. The molecules signal through cell surface receptors to activate intracellular signaling pathways that lead to the synthesis of contractile proteins and the transcription of matrix macromolecules [70]. Specific associations between multiple growth factors (FGFs, Wnt, Shh, PDGF, and others) and the matrix and their capacity to locally stimulate adherent cells are the keys to understanding the differentiation/pathological state of the cardiac environment [71,72]. The idea is that administering growth factors early after an infarction can stimulate angiogenesis. In mammals, these newly formed vessels improve blood flow to ischemic areas—that is, regions starved of oxygen—thereby rescuing potentially salvageable heart tissue. This approach leverages the concept that early intervention can limit the extent of infarction damage [73,74]. Conversely, in zebrafish, stimulating the trans-differentiation of epicardial, endocardial, and myofibroblast cells reveals underlying control mechanisms that could be harnessed in mammals [5,8,75]. Researchers hope to translate these insights into therapeutic strategies for human heart repair. This dual strategy reflects the different biological mechanisms in varied models (mammalian versus zebrafish systems), offering insights into how regenerative medicine might be tailored across species or even inspire innovative hybrid approaches in human therapies.

Fibroblast growth factor (FGF) can fall under many different subtypes and is produced by cells of mesenchymal origin [76]. The signal pathways mediated by FGF are the RAS/MAP kinase pathway, PI3 kinase/AKT pathway, and PLCγ pathway, among which the RAS/MAP kinase pathway is known to be predominant [80,132]. In the vertebrate heart, FGF is produced during development and in regeneration, and some aspects are also produced during hypertrophy by the pericardium, epicardium, endocardium, endothelium, and especially by fibroblasts of types FGF1/2 (basic), FGF4, FGF8, and FGF17b. FGFs stimulate proliferation, the emission of growth factors, and the migration of endothelial cells and fibroblasts, promoting the formation of new blood vessels and the reorganization of the ECM. Among them, FGF1 is sometimes referred to as the ‘universal ligand’, as it is capable of activating all seven different FGFRs; additionally, FGF17b and receptors FGFR (fibroblast growth factor receptor) 2 and 4 play particularly important roles in the developing heart and, together with another growth factor, namely PDGF-BB (platelet-derived growth factor-BB), they guide cardiac regeneration by resuming the embryonic cardiac development program [5,50].

Traditionally, pleiotropic transforming growth factor-β1 (TGF-β1) was seen as the central modulator of fibrosis in the heart, primarily because of its powerful effect on ECM production and fibroblast activation. However, our current understanding recognizes that its regulation is multifaceted and context-dependent. It is secreted by the epithelium (i.e., epicardium, endocardium, and endothelium) on stimulation of protease release (e.g., matrix metalloproteinase (MMP)-2 and -9), thrombospondin-1 binding, reactive oxygen species, pH extremes that denature LAP, integrin binding, and the mechanical separation of LTBP and LAP on stiff matrices [133,134]. While TGF-β1 remains a key player in fibrosis, research has expanded our perspective to see it as part of a broader network of signaling pathways. This network includes other cytokines, growth factors, and mechanical cues that together orchestrate the fibrotic response [133,134]. For example, excessive TGF-β1 activity can result in pathological hypertrophic scars due to overproduction of ECM components during the healing process. The intricate interplay among these factors explains why therapeutic strategies now often target multiple pathways simultaneously rather than focusing solely on TGF-β1.

Wnt signaling is fundamental in the developmental control pathway in the hearts of vertebrates [135]. Wnts are heterogenic proteins of about 40 kDa in size and are rich in cysteines; during their synthesis, they are modified by the attachment of a lipid, an acyl group termed palmitoleic acid [136,137]. Wnt/beta-catenin signaling acts bi-phasically in cardiac tissue, either promoting or inhibiting cardio-genesis depending on timing in mammals [138] as well as in zebrafish [37]. At early developmental stages, it promotes cardio-genesis, and it inhibits it later when the tissue undergoes differentiation. In zebrafish, Wnt/β-catenin signaling is active in the injured heart, where it induces fibrosis, promotes transdifferentiation, and prevents cardiomyocyte cell cycling [39]. Wnt types and Wnt antagonists are induced in the epicardium, endocardium, and in myofibroblasts during cardiac development and injury repair [39]. Five canonical/non-canonical routes are involved in Wnt pathways; these involve not only Wnt/β-catenin, but also the activation of G-protein/PLC/PiP2 and calcium release and the activation of the LRP-coreceptor in Rock/Rho signaling [40]. Understanding of the roles and mechanisms of the Wnt signaling pathway in injured zebrafish hearts can contribute to therapeutic strategies for human diseased hearts.

Neuregulin-1 (NRG-1) is a growth factor that is secreted in a paracrine manner or activated via a juxtacrine mechanism by endothelial and/or endocardial cells [139,140]. NRG-1 consists of more than 31 isoforms produced by alternative splicing [140]. The mature molecule has an EGF-like domain consisting of three regions, essential for binding to ErbB receptors. In particular, the juxta-membrane region serves as the proteolytic cleaving site for matrix metalloproteinases. The β-isoform of the molecule, obtained by proteinase cleaving, is required for cardiac development and in cardiovascular disease [141]. In post-natal mammals, and limited to this age group, NRG-1 stimulates DNA synthesis and postnatal cardiomyocyte proliferation of differentiated cardiomyocytes that undergo sarcomere disassembly and trans-differentiation of the cells [83]. For this reason, it has been implicated as a central regulator of both hypertrophic and hyperplastic cardiomyocyte growth. The NRG-1/ERBB pathway has been shown to activate PI3K/AKT, MAPK, and SRC/FAK pathways in cardiomyocytes, but it can also activate other cardiac cell populations [140]. In zebrafish, NRG-1/ErbB signaling controls the dialogue between macrophages and neural crest-derived cells during development and regeneration [83,141]. The possibility of using NRG-1 as a therapeutic molecule has been demonstrated by some studies. NRG-1 attenuated right ventricular hypertrophy, fibrosis, and failure in a mammalian pulmonary hypertension model, which may have been either direct or secondary to the effects on pulmonary vascular resistance [139]. Moreover, it can indirectly or directly stimulate macrophages in cytokine release, improving the protective and regenerative profile in pathophysiological conditions. In injured zebrafish, the reactivation of NRG-1 expression induces cardiomyocyte dedifferentiation, overt muscle hyperplasia, epicardial activation, and increased vascularization and causes cardiomegaly through the persistent addition of wall myocardium [85].

Platelet-derived growth factor (PDGF) is a protein expression of *pdgf-a* and *pdgf-b* genes, shared by mammals and zebrafish [77]. When the PDGF ligand binds to its receptor, which is a receptor tyrosine kinase, the receptor dimerizes. This dimerization brings the intracellular kinase domains of each receptor into close proximity, leading to trans auto-phosphorylation on specific tyrosine residues. These phospho-tyrosine sites then act as high-affinity docking points for downstream signaling molecules, resulting in the generation of diacylglycerol (DAG) and IP3 with subsequent release of calcium from the intracellular compartments [78]. PDGF signaling exerts mechanical and biochemical influences on fibroblasts within the ECM; it contributes to altering the mechanical properties and organization of ECM fibers. This stress may trigger fibroblasts to change their behavior, potentially promoting the secretion of different ECM components or remodeling enzymes that soften or restructure the matrix [127]. Such changes can be crucial for creating an environment that facilitates cell migration, vessel formation, and ultimately, heart tissue regeneration. Chemical inhibition of the PDGF receptor decreases the DNA synthesis of cardiomyocytes both in vitro and in vivo during regeneration [77] and can also promote the survival of epicardial and endocardial cells in vitro and regeneration in zebrafish hearts cultivated ex vivo [5].

The Sonic Hedgehog (Shh) signalling pathway is involved in the regulation of multiple aspects of heart development as well as the promotion of cardiomyocyte formation [33]. In the zebrafish model, Shh signalling was found to be activated during cardiac regeneration [79], and it may be fundamental in activating regeneration in post-natal mice [36].

Insulin-like growth factors (IGFs) are important growth factors in the development of fish and mammal hearts. They act on a complex consisting of cell-surface receptors (IGF1R and IGF2R), two ligands (IGF-1 and IGF-2), and IGF-binding proteins (IGFBP) [81]. IGF2 has been shown to be important for cardiomyocyte proliferation and heart growth during mid-gestation heart development in mice, and it is an important requirement for activating GATA4+ cardiomyocytes in zebrafish [81,82].

### 3.5. Interleukins and Other Inducing Chemicals

The interconnection between inflammation and heart failure is mutually reinforced during the regeneration process. Ischemic cardiac injury in mammals as well as in adult zebrafish can be induced by hypoxia/reoxygenation; consequently, it provokes oxidative stress, inflammation, death, and the proliferation of cardiomyocytes [142,143]. IL-18, a myocardial proinflammatory cytokine that is particularly elevated in hypertrophic hearts, is released by cells via the canonical pyroptosis pathway. IL-18 seems to be related to induction of the activation of hypertrophy genes in subsequent myocardial cells, since no signs of hypertrophy-related genes have been observed in IL-8 KO mice [144]. Fibroblasts and macrophages produce IL-6 after myocardial infarction; in an appropriate quantity, IL-6 effectively ameliorates adverse cardiac remodeling and prevents cardiac dysfunction due to myocardial infarction [145,146].

## 4. MicroRNAs and ECM Exosomes Play a Role in Cardiac Cell Differentiation and Pathology

Small regulatory non-coding RNA molecules include microRNAs (miRNAs) and small interfering RNAs (siRNAs), which are united by their association with the Argonaute family of proteins and by their function [12]. MicroRNAs are phylogenetically conserved and regulate the translation of target messenger RNAs, providing a mechanism for protein-dose regulation [147]. These miRs are produced by cells during development, differentiation, and adulthood; however, their overexpression or, conversely, downregulation can drive cells toward de-differentiation, proliferation, or pathological states. Thus, the fate of the tissue homeostasis is strictly linked with the growth factors that come from the ECM and the balance of miRs present in the cytoplasm. Previous research has shown that miRNAs use extracellular vesicles (exosomes) as carriers to achieve cell-to-cell communication through the ECM [148]. It has been suggested that following myocardial infarction, exosomes released from parenchymal cells (endocardial, endothelial, and epicardial cells) can carry miRs [149]. Exosomes are lipid bilayer biological nanovesicles of 30–150 nm in diameter, loaded with various proteins, lipids, and mRNAs/miRNAs [149]. Released exosomes spread in the form of paracrine or remote secretion and are then adsorbed by donor cells. The exosome contents form an Argonaute/miRISC complex and combine with the mRNA of the target gene based on partial sequence complementarity (about 7 to 15 nucleotides) [149,150]. A large body of miRs has been identified as contributing to cardiac development, regeneration, and disease, and several of these are shared between zebrafish and mammals [8,151] (Figure 1). In the healthy mammalian heart, many miRNAs are highly expressed in cardiac tissue and thus play a key role in cardiac homeostasis [151]. These include miR-1, miR-16, miR-27b, miR-30d, miR-126, miR-133, miR-143, miR196b, miR-208, and the let-7 family [152]. In zebrafish, similar miRs have been reported, such as miR-1, miR-133a/b, miR196b, and miR-375/notch; miR499 prevents cardiomyocyte apoptosis by suppressing the calcineurin-mediated dephosphorylation of dynamin-related protein-1 [11,153,154]. miR-19a is induced by Tbx5, and a defined dosage of miR-19a is essential for the correct development of the heart; the TBX5 protein is, in turn, the target of miR182-5p [155]. miR-29c regulates the lateral development and cardiac circulation of zebrafish embryos by targeting Wnt4 growth factor [156]. In acute ischemic or infarction conditions that activate regeneration, as well as in chronic hypertrophy, a large set of miRs are expressed similarly, mirroring their patterns during development (rev. [151,152]). Comparably, in zebrafish, a multitude of similar miRNAs are expressed under similar conditions, where the key miRs are miR-1, miR-1331a, miR-133b, miR196b, miR30c, and miR-499 [11,47,157,158,159,160,161]. In particular, miRs-1 and -133a seem to be related to the FGF pathways, whereas miR-133b is related to endothelium/NFAT2 regulation [8,11,65]. Thus, the zebrafish seems to be a promising model to study the miR pathway in mammalian heart development and disease.

## 5. Mechanical ECM Pressure Can Provoke Cardiac Pathologies

Pathological cardiac hypertrophy is connected to processes such as autophagy, oxidative stress, and the death of cells. It is characterized by the enlargement of individual cardiomyocytes, the ensuing progression of myocardial interstitial fibrosis, and a decrease in myocardial ventricular chamber dilation, resulting in systolic–diastolic dysfunction, leading to heart failure and arrhythmia [10]. New fields of research have focused on the mechanical influence on fibrosis and cardiac pathologies. The mechanical information transmitted by the ECM is largely processed through integrin-based adhesions, which provide a mechanical link between the matrix and the intracellular cytoskeleton [87]. These interactions allow the activation of transcription pathways that are capable of, among other things, remodeling the cytoskeleton itself. Remodeling of the cytoskeleton can transmit forces to the nuclear lamina, the nucleoskeleton, potentially altering the environment for transcription. Mechanosensitive proteins, such as nesprin and the lamins that form the LINC complex, can directly interact with and control chromatin organization, limiting or promoting the accessibility of transcription factors to specific genetic loci [162].

Alterations in cytoskeletal tension appear to be correlated with the acetylation state of histones H3 and H4, and therefore, with the recognition of heterochromatin remodelling complexes so that these become transcriptionally active [163]. Compared with soft substrates, the increased traction forces that occur on rigid substrates alter nuclear stress receptors and, in a cascade, the transcription of genes involved in cell proliferation and differentiation [162,163,164,165]. The perpetuation of the activated state of fibroblasts associated with infarction scarrng could therefore modulate the transcriptional activity of genes involved in the progression of fibrosis [165,166,167]. A growing body of evidence supports the idea that cells not only respond transiently to mechanical signals but also “memorize” the mechanical information present in the local microenvironment and integrate that information through epigenetic mechanisms. It has been proposed that the mechanical memory retained by fibroblasts may have long-term effects on cells, influencing their fate [168,169,170]. However, no epigenetic drugs other than statins, which interfere with calcium signaling, have currently been developed for the treatment of fibrotic diseases.

Even mechanical pressure due to pressure overload can induce pathology by inducing apoptosis. Apoptosis plays a significant role in cardiovascular diseases such as atherosclerosis, ischemic heart disease, cardiac hypertrophy, congestive heart failure, and myocardial remodeling. It has been demonstrated that oxidative/physiological stress and inflammatory compounds (cytokines, tumour necrosis factor, and Fas ligand) can induce cardiovascular system apoptosis. Once apoptosis has been activated, dead cells are replaced by an ECM, which in turn damages cardiomyocytes and increases interstitial fibrosis, thus promoting myocardial hypertrophy [171]. Apoptotic mechanisms, such as the extrinsic, intrinsic, or granzyme pathways, have recently been identified as occurring in three separate activation types. Another type of activation has also been described—pyroptosis, which is able to activate the same caspases in the extrinsic and intrinsic pathways. In the extrinsic pathway, signal molecules known as ligands, which are released by other cells, bind to transmembrane death receptors on target cells to induce apoptosis by recruiting caspase-8 (an apoptosis activator) to the cytoplasm. Active caspase-8 can also cleave BID, which results in the release of cytochrome c (together with BAX and BAK) in mitochondria via the intrinsic pathway. Following release, cytochrome c forms a complex in the cytoplasm with adenosine triphosphate (ATP), an energy molecule, and Apaf-1 (an enzyme). Following its formation, this complex activates caspase-9. This latter works together with the complex of cytochrome c, ATP, and Apaf-1 to form an apoptosome, which in turn activates caspase-3, the effector protein that initiates degradation [172,173].

Granzyme B can engage the death pathway at multiple entry points [173,174]. The granzyme B/perforin system is affected mainly by circulating cytotoxic T lymphocytes and natural killer cells (nonspecific cells in fish [175]), which contain secretory granules released at the inflammation site via cell-to-cell contact, the engagement of integrin with the ECM, and cytokines and chemokines [176]. Perforin provokes transmembrane channels and allows granzyme B access into target cells. Granzyme is a serine protease that can activate caspase 8 and degrade structural proteins in the BID cascade. Thus, granzyme B activation can activate both extrinsic and intrinsic pathways. Extrinsic and granzyme mechanisms, as well as inflammation, can both also be activated by mechanical stimuli and toxic substances [173,176,177].

Pyroptosis, a regulated cell death pathway driven mainly by the activation of caspase 1, has recently been shown to play an important role in heart pathology. Pyroptosis is activated by strong mechanical compression in cells close to dead cells (in the infarcted area). Damage-associated molecular patterns (DAMPs) are released from damaged myocardial cells to activate the formation of an apoptosis-associated speck-like protein containing a CARD (ASC) that interacts with NACHT, LRR, and PYD domain-containing protein 3 (NLRP3), resulting in caspase-1 cleavage and its relative activation [127]. This process results in the entry of the cell into extrinsic or intrinsic apoptotic mechanisms. In addition to apoptosis and pyroptosis, a new emergent mechanism called necroptosis, which can cause cell death when caspase-8 activity is inhibited, has been described [176,177]. The death receptor signaling pathway activates the RIPK3 and RIPK1 complex, which can interact with the mixed lineage kinase domain-like protein (MLKL)-induced RIPK1/RIPK3/MLKL complex [178,179]. This complex activates necroptosis, which can cause a widespread inflammatory response through the release of endogenous molecules, the disruption of cell membranes, and the disintegration of organelles, with no obvious morphological changes in the chromatin of the nucleus [178]. Another process activated by mechanical stress is pathological autophagy [179]. This process removes damaged mitochondria accumulated in the hypertrophied myocardium, thereby reducing ROS production in cardiomyocytes. However, when this process is extensive, it can disrupt the metabolic equilibrium in cardiomyocytes. The mechanical induction of autophagy is widespread across species and those of the TOR-independent type, involving the EGFP-LC3 or ATG5 molecule in vertebrates [180]. Mechanic compression can also increase calcium-sensing receptor (CaSR) expression and induce autophagy in hypertrophied hearts [181].

## 6. Managing the Matrix Could Be a Therapeutic Strategy: Next-Generation Engineered Connective Tissue Designed to Effectively Deliver Chemicals and Growth Factors

Therapeutic strategies target the epigenetic mechanism of pathological cells, aiming to efficiently and definitively “erase” their persistent fibrogenesis memory and reestablish their quiescent state. Exosomes can be exploited for therapeutic applications, including gene therapy and/or anti-fibrosis therapy. They can contain materials with diverse characteristics and are designed to facilitate cellular uptake through precise molecular interactions [182,183]. Their capacity for homotypic targeting and self-recognition provides opportunities for personalized medicine [184]. The recent discoveries of epicardial cells’ heterogeneity during the development/differentiation of cardiac tissue, and the possibility of driving some epicardioid cells to differentiate into muscle instead of fibroblast or pericytes, open new directions of research using combinations of growth factors, exosomes, and ECM [184]. The regenerative potential of growth factors has gained significant importance due to stem cell administration and the implantation of growth-factor-enriched ECM. These interventions support trans-differentiation, antiapoptotic, and angiogenic effects on resident cells, potentially reducing inflammation and fibrosis in infarcted areas [149,185,186]. In this context, reviews on the role of the extracellular matrix (ECM) and growth factors in mammalian heart development and regenerative stimulation can provide valuable insights [38,187,188]. The possibility of using ECM implants in stroked hearts could be a way of curing the post-infarction fibrosis that occurs in mammals. The phosphorylated form of 7-amino-acid-peptide (7A) domain (encoded by a short open-reading frame of the HDAC7 gene e) can promote the regeneration of cultured cardiac tissue. Moreover, in vivo infarcted mice models demonstrated that the intra-myocardial delivery of 7Ap-loaded collagen hydrogel promoted neovascularization, stimulated stem cell recruitment and differentiation, reduced cardiomyocyte apoptosis, and promoted cell cycle progression [189]. The field of tissue engineering has focused on developing biological scaffolds of ECM to promote cell growth and tissue regeneration at the wound site, either ex vivo or in vitro [5,189]. Hydrogels, formed through the physical and chemical crosslinking of water-soluble polymers and collagens, have been studied as potential scaffolds [188,189,190,191,192,193,194,195]. Even the use of Integra (IntegraLife), composed of a collagen and chondroitin-6-sulfate matrix, and Culturex (Trevigen), which includes basement membrane and ECM compounds, has shown positive results in dermal models and zebrafish hearts ([195]; Romano et al., personal communication). These materials facilitate cell proliferation and the efficient delivery of substances throughout the scaffold network. Establishing appropriate extracellular matrix (ECM) support in the future will be crucial for effective heart regeneration.

## Figures and Tables

**Figure 1 cells-14-00875-f001:**
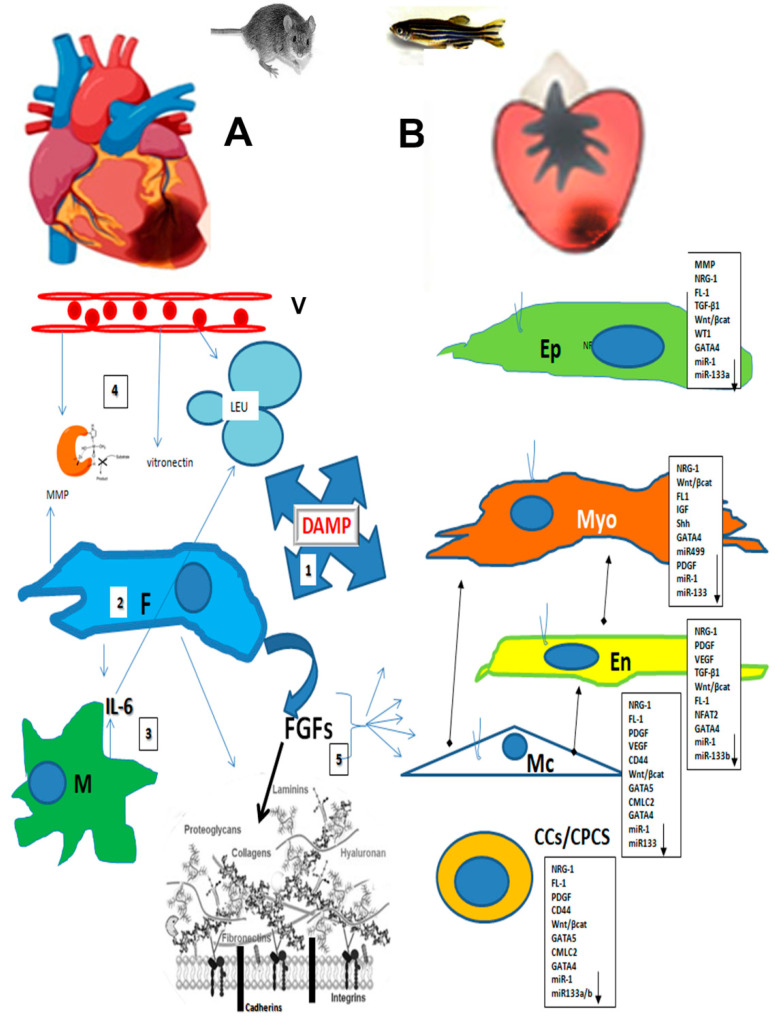
ECM’s role in inducing cardiac components. After ischemic damage, the first action in the site is alarmin emission (1) by endothelial cells of the capillary (V) and fibroblasts of the ECM (2). DAMP molecules activate apoptosis but also stimulate fibroblasts to produce FGFs. In parallel (3), fibroblasts and resident macrophages (M) begin producing IL-6, which stimulates the migration of leukocytes (LEU) to the damaged site. Simultaneously (4), the release of platelets, vitronectin, and metalloproteinases (MMP) takes place, which act to form a clot and manage the ECM structure. Fibroblasts are stimulated to produce new connective matrices like fibronectin, HA, laminins, GAG, and collagens. Fibronectin and GAGs are particularly involved in FGF release and other growth factors among the heart components. (5) The cellular heart components—epicardium (Ep), myocardium (Myo), endocardium (En), mural cells/pericytes (Mc), and cardiac stem/progenitor cells (CCs/CPCs)—are activated by several growth factors and by integrin/receptors/cadherins and, in turn, start to regulate key gene expressions that cause cells to de-differentiate and/or proliferate to repair the damage site (panels). The key gene expression is comparable between mammals (**A**) and fish (**B**); however, the speed of cellular replacement in mammals is low compared with that in fish, and a large amount of ECM is secreted by fibroblasts in the process of recovering the damaged tissue, provoking a fibrotic scar.

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
