# Peer review of "The Role of the Extracellular Matrix in Inducing Cardiac Cell Regeneration and Differentiation"

_cells, 2025, doi:10.3390/cells14120875_

Round 1

Reviewer 1 Report

Comments and Suggestions for Authors

Unfortunately the clarity of this review is obscured by the inaccurate use of language. Some can be attributed to "lost in translation", but others are clearly wrong. For example, line 210, ."...type III collagen has alpha 1 and two alpha 2 chains longer than other alpha chains collagen types." Type III collagen is a homotrimer with three a1 chains and the phrase "longer than other alpha chains collagen types has no meaning. There are over 20 different types of collagen, to what other alpha chain collagen types is the author referring to? This is just one example but there are many. In addition, references provided for cited work do not align with the content of the paper. 

Comments on the Quality of English Language

Poor, to a degree to which the review is difficult to understand or to gain pertinent information from.  

Author Response

Thank you to the Referee for the comments, I have worked on the text, and I improve the clarity in several setions.

Review 1: Unfortunately the clarity of this review is obscured by the inaccurate use of language. Some can be attributed to "lost in translation", but others are clearly wrong. For example, line 210, ."...type III collagen has alpha 1 and two alpha 2 chains longer than other alpha chains collagen types." Type III collagen is a homotrimer with three a1 chains and the phrase "longer than other alpha chains collagen types has no meaning. There are over 20 different types of collagen, to what other alpha chain collagen types is the author referring to? This is just one example but there are many. In addition, references provided for cited work do not align with the content of the paper.

“Collagen is composed of repeating units of tropocollagen, a protein complex with a molecular mass of approximately 285 kDa, consisting of three α polypeptide chains. These α chains are encoded by 30 genes located across at least 12 different chromosomes, exhibiting slight variations in their amino acid sequences. This results in five distinct chains (labelled α1, α2, α3, α4, α5) that can variably associate to form a triple helix through hydrogen bonds. There are over 20 types of collagen, with variations arising from the assembly of tropocollagen monomers and the composition of the three α chains. Type I collagen consists of two α1 chains and one α2 chain with slightly different compositions, while type III collagen comprises one α1 chain and two α2 chains [95]. Additionally, type IV collagen is formed by the assembly of five different α chains and is anchored to the basement membranes of myocardial cells, making it essential for the elasticity of the contractile system [96].”

R1_ Poor, to a degree to which the review is difficult to understand or to gain pertinent information from. 

The English style has been revised by the MDPI English supporting system

Reviewer 2 Report

Comments and Suggestions for Authors

In the review the author explores different pathways where ECM can play a role in modulating cardiac cell fate.

Main comments:

The author should clarify how does this review add to the current view of this field. Moreover, various previous reviews have explored the role of ECM in heart development and disease (e.g. PMID: 21618406, PMID: 33674261, PMID: 28459429). It should be further clarified why is it relevant to compare human and zebrafish in this perspective.

The topic “2. Inducing Cardiac Cells by Key Morphogenic Factors” mentions some pathways and key transcription factors but is not clear the choice of such examples. There is also a lack of a clear flow in the cardiac development. An illustrative figure would be an important added value.

The topic “3. Inducing Cardiac Cells by Key Extracellular Components” is not clearly structured. The highlighted molecules are not all ECM components, at the most they are part of the matrisome. Some ECM components known to play important role in cardiac development are not detailed (laminins, fibronectin, collagens), and is not clear why.

Lines 55-58: Is not clear why the therapeutics using human iPSCs and in zebrafish are contradictory. They are two different approaches. Moreover, if the therapy with iPSCs showed partial success, it should be further detailed why would therapies that do not involve stem cells be to some extent preferred.

Figure 1 is not clear. Are the genes/pathways common in humans and zebrafish? This is not enough explored in the text and the figure is not clear.

Specific comments:

The acronym “ECM” should be used throughout the text from the moment it is defined. This should be revised.

Line 20: The author should provide some examples of the 18 risk factors

Line 44: The author should consider replacing “cardiosphere” by “cardiospheres”

Line 56: Should be “have been injected”

Line 69: “exocytosis morphogenetic factors” are terms that do not relate to the pathways referred to. Not clear what the author wants to refer with “exocytosis”

Line 76-78: Rephrase. The word “form” is extensively used in the sentence.

Line 94: Revise the word “couls”

Line 97: “alarmins” defined a much broader class of molecules. This sentence should be re-phrased to convey this idea

Line 123: should be “have been localized”

Line 136-137: The sentence needs to be rephrased. Is not clear. Meaning of “specific niches” should be clarified.

Lines 146-150: The sentence needs to be rephrased. Is not clear that there are different alpha and beta integrin chains that can have distinct combinations.

Comments on the Quality of English Language

English must be revised. I highlighted some examples, but are many throughout the text.

Author Response

I am grateful to Referee 2 for the comments that have certainly improved the manuscript. Thus, the text has been polished to better expose the aims of the review and enhance clarity by also inserting a new table (Table 1).

R2_ The author should clarify how does this review add to the current view of this field. Moreover, various previous reviews have explored the role of ECM in heart development and disease (e.g. PMID: 21618406, PMID: 33674261, PMID: 28459429). It should be further clarified why is it relevant to compare human and zebrafish in this perspective.

The review focuses on the common effects of ECM components, mechanical induction and relased growth factors in zebrafish vs mammals. This approach is original compared to previous manuscripts. I have added the following sentence in the introduction: “This review analyzes key factors in the ECM that are involved in heart regeneration and differentiation, which are common in both zebrafish and other models such as mammals. These insights could serve as a foundation for developing new research strategies for heart repair.” Additionally, I have incorporated references to other reviews in the last paragraph as important complements to the knowledge regarding mammals.

R2_The topic “2. Inducing Cardiac Cells by Key Morphogenic Factors” mentions some pathways and key transcription factors but is not clear the choice of such examples. There is also a lack of a clear flow in the cardiac development. An illustrative figure would be an important added value.

The key morphogenetic factors cited are related to the commonalities between zebrafish and mammals. Moreover, their action are similar because they induce the same embryonic genes. A complete panel of the induction and activity during cardiac development has been shown in another review present in this issue (Angom et al., 2025, 38). According to other authors, we decided to subdivide the topics. Thus, I am focusing the manuscript on the regeneration/role of ECM to avoid overlap. Anyway, I have inserted a new table (Table 1) summarising the common inducing factors between mammals and zebrafish.

R2_The topic “3. Inducing Cardiac Cells by Key Extracellular Components” is not clearly structured. The highlighted molecules are not all ECM components, at the most they are part of the matrisome. Some ECM components known to play important role in cardiac development are not detailed (laminins, fibronectin, collagens), and is not clear why.

This note coincides with those similar to R3. I have divided paragraph 3 into several sections that represent the different factors of ECM capable of directing cell fate. I have created two supplementary sub-paragraphs. Paragraph 3 now comprises five sub-paragraphs: 3.1. Cardiac cells and their basal lamina/ECM adhesions; 3.2. The basal lamina and ECM components; 3.3. Roles of hyaluronic acid and GAG; 3.4. Growth factors in the cardiac stroma; 3.5. Interleukins and other inducing chemicals;

R2: Lines 55-58: Is not clear why the therapeutics using human iPSCs and in zebrafish are contradictory. They are two different approaches. Moreover, if the therapy with iPSCs showed partial success, it should be further detailed why would therapies that do not involve stem cells be to some extent preferred.

I have better defined the sentences in lines 55-59 as follow: “Recently, heart muscle cells differentiated in vitro from induced pluripotent stem cells (iPSCs) have been successfully injected into a patient; however, partial repair has occurred due to the limited proliferation of these cells in situ [21]. In contrast, the zebrafish model has demonstrated a high capacity for proliferation and regeneration of resident cardiomyocytes, both in vivo and in ex vivo culture, without the need for stem cell injection [5,22]. Moreover, it was demonstrated that it is possible to regenerate a heart in ex vivo without the fish body, by using the right cocktail of growth factors [5].”

R2_Figure 1 is not clear. Are the genes/pathways common in humans and zebrafish? This is not enough explored in the text and the figure is not clear.

I have change the Figure 1 for better understand the shared genes between humans and zebrafish and the different genes in humans

R2_The Acronym “ECM” should be used throughout the text from the moment it is defined. This should be revised.

I have duly revised the text to use ECM acronym properly.

R2_Line 20: The author should provide some examples of the 18 risk factors

I have inserted some of the risck factors: ischemic heart disease and stroke, rheumatic heart disease, hyperthension etc)

R2_ Line 44: The author should consider replacing “cardiosphere” by “cardiospheres”

It has done

R2_ Line 56: Should be “have been injected”

The sentence has been changed accordly the R2 previous suggestion of line 58.

R2_ Line 69: “exocytosis morphogenetic factors” are terms that do not relate to the pathways referred to. Not clear what the author wants to refer with “exocytosis”.

The sentence has been changed in: The morphogenesis of the heart and the maintenance of its physiology require the activation and coordination of different transcriptional programs, which are activated starting from the expression of homeobox genes and by the signalling mediated by Nodal, BMP, FGFs, Wnt and Notch morphogenetic factors.

R2_Line 76-78: Rephrase. The word “form” is extensively used in the sentence.

The paragraph has been rephrased in: In zebrafish, following its development, the transient cells of the epicardium flatten and cover the forming heart to establish a contiguous of two to three cell layers

R2- Line 94: Revise the word “couls

It has been done, but the English correction has changed the sentence.

R2_Line 97: “alarmins” defined a much broader class of molecules. This sentence should be re-phrased to convey this idea

I have changed in: “In the ECM, these molecules are included in  "alarmins" category and, in particular, in the  DAMP typology (damage-associated molecular pattern). DAMP  are endogenous proteins released from damaged or dying cells that can also activate the inflammatory system and the machinery of heart regeneration in zebrafish as well as in mammals [58]”.

R2_Line 123: should be “have been localized”

The sentence has the meaning that the key embryonic genes for the developing/regenerating heart are in relation of the alternative translation system. This system is related to ancient homebox genes (3IF3d cap or IRES sequences).

The sentence has been changed in: I”nterestingly, the Wt1, NFAT2 and GATA4 transcription factors could be translated in an alternative translation process involving the RACK1 protein [65].”

R2_Line 136-137: The sentence needs to be rephrased. Is not clear. Meaning of “specific niches” should be clarified.

The sentence has been enhanced: “Stem cell resident cells in specific niches have been demonstrated in the heart and are fundamental to maintain cardiac homeostasis and myocardial repair following injury [68]. These niches are located in specific areas (close to supporting cells/pericytes and endocardium), representing specialised microdomains where the quiescent and activated state of resident stem cells is regulated [68].”

R2_Lines 146-150: The sentence needs to be rephrased. Is not clear that there are different alpha and beta integrin chains that can have distinct combinations.

The sentence has been re-phrased: “The consequence of this adhesion is the establishment of a large molecular network among the components of basal lamina (fibronectin, collagens of I and IV types, entactin, etc.) capable of inducing signal transduction in the cell. For example, under the adhesion stimulus, the transduction signalling can involve the RAS_ERK or RHO systems, which can bring the cell to expose new membrane molecules such as cadherins or other migrating integrins [69]. The integrins consist of two chains of α and β types. However, each chain can be variable in composition, depending on the domain's constitution. The α chain contains several domains in which the αA domain present in turn four subunits (α1, α2, α10, and α11) that are specific for linking the β chain. The β1 forms a distinct laminin/collagen-binding subfamily [70,71].”

R2_English must be revised. I highlighted some examples, but are many throughout the text.

The author have send the manuscript to the MDPI English revision system

Reviewer 3 Report

Comments and Suggestions for Authors

The review by Nicla Romano, “The Extracellular Matrix's Role in Inducing Cardiac Cell Fate” delivers a good and up-to-date review of the field. Before final publication I would like to raise some comments concerning visualization of the specific paragraphs and some minor points.

Major Points:

  1. Generally, I think the review is already structured well. However, I found that certain parts would benefit if the conclusion from the text would be shown in a table. This specifically concerns the paragraph 3.1.. This paragraph describes the different signaling factors such as FGF, TGFB1, WNT signaling, etc.. I think it would be beneficial for the reader if these factors could be summarized in a table that would include e.g.: Signaling factor, secreted by tissue or specific cell type, pathways involved, etc.
  2. Additionally, I think the manuscript would benefit if paragraph 3 is further subdivided into paragraphs specifically focused on the ECM protein introduced: Integrins, Cadherins, Fibronectin, etc., instead of one continuous text.

Minor Points:

  1. Please add the following references since they are crucial to some of the points raised in the manuscript:
    1. Jebran, A.-F., Seidler, T., Tiburcy, M., Daskalaki, M., Kutschka, I., Fujita, B., … Zimmermann, W.-H. (2025). Engineered heart muscle allografts for heart repair in primates and humans. Nature, 1–9. Retrieved 1 February 2025 from
    2. Meier, A. B., Zawada, D., De Angelis, M. T., Martens, L. D., Santamaria, G., Zengerle, S., … Moretti, A. (2023). Epicardioid single-cell genomics uncovers principles of human epicardium biology in heart development and disease. Nature Biotechnology. Retrieved from https://doi.org/10.1038/s41587-023-01718-7
  2. Please review the following text passages for language, meaning and sense:
    1. Line 27 “stromal”
    2. Line 61 “far to recover”
    3. Line 77-78 “forming heart to form”
    4. Line 90 “re-express”
    5. Line 123 “has”
    6. Line 141 “fundamental”
    7. Line 234 – Rephrase
    8. Line 244 “role”
    9. Line 514 “Mechanic”
    10. Line 530 “stroked”

Author Response

I wish to give particular thanks to this referee for the precious suggestions that I have duly followed.

R3_ Major Points: Generally, I think the review is already structured well. However, I found that certain parts would benefit if the conclusion from the text would be shown in a table. This specifically concerns the paragraph 3.1.. This paragraph describes the different signaling factors such as FGF, TGFB1, WNT signaling, etc.. I think it would be beneficial for the reader if these factors could be summarized in a table that would include e.g.: Signaling factor, secreted by tissue or specific cell type, pathways involved, etc.

I have inserted a Table that summarized the growth factors and their action, as Referee n.3 has suggested.

R3_Additionally, I think the manuscript would benefit if paragraph 3 is further subdivided into paragraphs specifically focused on the ECM protein introduced: Integrins, Cadherins, Fibronectin, etc., instead of one continuous text.

I have done two sub paragraphs. The 3 paragraph is now composed by 5 sub-paragraph: 3.1. Cardiac cell and their basal lamia/ECM adhesions; 3.2 The basal lamina and ECM components; 3.3. Roles of hyaluronan acid and GAG; 3.4. Growth Factors in the Cardiac Mood; 3.5. Interleukins and Other Inducing Chemicals;

R3_Please add the following references since they are crucial to some of the points raised in the manuscript:

I have read and inserted the references suggested by the referee.

R3_Please review the following text passages for language, meaning and sense.

I have done all corrections, I have used the MDPI English supporting system.
